# Artificial intelligence for research capacity strengthening: Two reviews and a pathway to shift power in global health

Brian Wahl[1]*, Tiffany Nassiri-Ansari[2], Daniel D. Redpath[3], Pascale Allotey[4], Nina Schwalbe[2,5]

1 Department of Epidemiology of Microbial Diseases, Yale School of Public Health, New Haven, Connecticut, United States of America, 2 Spark Street Advisors, New York, New York, United States of America, 3 Institute for Global Health and Development, Queen Margaret University, Edinburgh, United Kingdom, 4 Department of Sexual and Reproductive Health and Research, which includes the UN Special Programme of Research, Development and Research Training in Human Reproduction, World Health Organization, Geneva, Switzerland, 5 Center for Global Health Policy and Politics, Georgetown University, Washington, D.C., United States of America

* brian.wahl@yale.edu

## Abstract

Significant disparities persist in how researchers from low- and middle-income countries (LMICs) and high-income countries (HICs) participate in agenda-setting and knowledge production. Rapid advancement in artificial intelligence (AI) might contribute to improving research capacity in LMICs. This review aimed to synthesize evidence on AI for research capacity strengthening in LMICs towards shifting power in global health. We conducted a systematic review of current evidence on AI for research capacity strengthening and a review of reviews on the decolonization of knowledge generation, searching PubMed, Scopus, and SciELO for relevant literature. Articles were included in the systematic review if they included primary data on using AI for research purposes. Reviews were included in the review of reviews if they addressed issues related to knowledge generation. Each review was assigned two independent reviewers for title and abstract screening, full-text review, and data extraction. A narrative synthesis of the extracted data from both reviews was then performed. Given study designs for the inclusion-eligible papers, we did not conduct a formal risk-of-bias assessment. The systematic review identified 305 papers, of which 8 met the inclusion criteria. The review of reviews identified 14 papers, of which 8 were included in the final analysis. Key themes identified from the systematic review include data analysis and research productivity, literature reviews and knowledge management, training and capacity strengthening, expanding access to methodological support, and writing support. The review of reviews found a recurrent theme in the need to address power imbalances rooted in colonial legacies. These reviews demonstrate the potential for AI to transform research capacity in LMICs by democratizing access to advanced analytical tools, providing methodological support,

**Data availability statement:** All data are in the manuscript and/or supporting information files.

**Funding:** This research was supported in part by funding from the Bill & Melinda Gates Foundation to BW. The funder had no role in study design, data collection and analysis, decision to publish, or preparation of the manuscript.

**Competing interests:** The authors have declared that no competing interests exist.

and helping overcome resource limitations that have historically restricted research opportunities. However, equitable governance and local leadership are crucial to prevent AI from widening the gap between LMICs and HICs, perpetuating the power asymmetries that current efforts seek to dismantle.

## Author summary

Global health research has been shaped by inequalities rooted in colonial legacies. Researchers in wealthier countries have, in general, led the establishment of research priorities and knowledge generation. Artificial intelligence tools are rapidly changing how research is done. We wanted to understand whether these technologies could help shift the power balance in global health by strengthening the ability of researchers in lower-resourced settings to lead research efforts. We conducted two complementary literature reviews: one examining how artificial intelligence is being used to build research skills and capacity, and another exploring efforts to decolonize knowledge production in global health. We found that artificial intelligence tools show promise in areas like data analysis, literature management, and scientific writing, potentially helping researchers overcome barriers related to limited infrastructure or expertise. However, the evidence base remains limited, and there are real risks that these technologies could deepen existing inequalities if developed without meaningful input from the communities they aim to serve. Our findings suggest that realizing the benefits of artificial intelligence for global health research depends on centering local leadership, equitable governance, and sustained investment in people and systems.

## Introduction

Equitable research capacity is fundamental to effective global health. However, significant disparities persist in how researchers from low- and middle-income countries (LMICs) and high-income countries (HICs) participate in agenda-setting and knowledge production [1]. Academic authorship patterns reflect these disparities, with researchers from high-income countries dominating lead positions in global health publications [2–5]. Researchers have identified colonial legacies and power asymmetries that perpetuate the dominance of Northern institutions in defining research priorities, directing funding, and setting the standards for what is considered "valid" knowledge [6].

Empowering LMIC researchers to shape and lead studies is essential for transforming global health and improving research outputs [7]. Research capacities are therefore both shaped by power asymmetries with colonial roots and capable of addressing those very asymmetries. However, strengthening research capacity does not in itself resolve the deeper structural inequities rooted in funding flows, agenda-setting power, and institutional dominance, and any capacity-focused intervention, AI included, must be understood within this broader political economy. The

momentum toward locally driven scholarship has fostered a broader dialogue on dismantling entrenched hierarchies and elevating diverse epistemologies in health research [8].

Concurrently, artificial intelligence (AI) has rapidly advanced, with emerging applications ranging from advanced data analytics to generative models [9]. Machine learning (ML) algorithms and large language models (LLMs) have expanded the tools available to researchers, helping them analyze complex datasets [10], make predictions about complex systems [11,12], and formulate research hypotheses [13].These new technologies, particularly LLMs, can improve research capacity in LMICs by making advanced analytics more accessible while reducing reliance on external expertise, consequently addressing contemporary imbalances in power dynamics between and among international research collaborations. For example, sexual and reproductive health and rights (SRHR) research faces many of the inequities in knowledge production and leadership that riddle global health more broadly. AI offers opportunities to enhance SRHR research capacity by streamlining data analysis, improving access to literature, and supporting the generation of evidence for policy interventions.

However, AI development and deployment remain unevenly distributed, and there remains a risk of reproducing existing power imbalances if the technology is not developed and governed in ways that prioritise equity [14,15]. Most research on using AI for global health has focused on improving care and supplementing human resource capacity through specific health applications [15,16].

In this review, we aim to examine how AI—including recent advances in generative AI—can strengthen research capacity in LMICs toward improving research outputs and shifting power in global health. We synthesize existing literature on AI-driven capacity-building initiatives and highlight examples of local scientists leveraging such technologies to address context-specific research questions. This review explores new approaches to making global health knowledge generation more inclusive and community-oriented. To integrate these perspectives, we paired a systematic review of AI-enabled capacity strengthening with a review of reviews on decolonization in knowledge production to ground our analysis of AI within the broader structural debates that shape global health research. The findings of this review could contribute to ongoing discussions on how equitable partnerships and local leadership, coupled with emerging digital tools, might create pathways toward decolonizing global health research.

## Methods

### Search strategy and selection criteria

We conducted two literature reviews in accordance with the Preferred Reporting Items for Systematic Reviews and Meta-Analyses (PRISMA) guidelines [17] and the Joanna Briggs Institute (JBI) methodology [18]. The first was a systematic review examining the current evidence related to using AI for research capacity strengthening and capacity strengthening for using AI in research. The second was a review of reviews assessing the published literature on the decolonization of knowledge generation and epistemology in global health.

For both reviews, we searched for English, French, Portuguese, and Spanish language papers published between January 1, 2000, and December 31, 2024, in two databases: PubMed and Scopus. We conducted additional searches in Portuguese and Spanish on SciELO to surface non-English literature. For the systematic review, we used a broad search strategy, including two themes: "research capacity" and "artificial intelligence." For the review of reviews, we used a narrower search strategy with four main themes in the languages specified above: "decolonization," "knowledge generation," "review articles," and "public health." The specific search terms for both reviews are included in S1 Text.

We used a two-stage screening process for both reviews using Covidence software for data management. After removing duplicates, two reviewers (BW and TNA for the systematic review and BW and TNA for the review of reviews) independently screened titles and abstracts using the inclusion criteria. Articles that passed initial screening underwent full-text review by the same two reviewers. For the systematic review, we included studies that included primary data on using AI for research purposes. We excluded studies that assessed the use of AI for supplementing skills unrelated to research

or used AI to evaluate research (e.g., bibliometric analyses). For the review of reviews, we excluded reviews that did not directly address issues related to knowledge generation. After screening, the same reviewer pairs independently extracted relevant data, themes, and examples from the included articles. All screening and data extraction conflicts were resolved through consensus discussion between the two reviewers.

## Data analysis

We then performed a narrative synthesis of the extracted data for both reviews, organizing findings into key themes that emerged during the data extraction process. Given the heterogeneity of studies from the systematic review and the small number of papers from the review of reviews, we did not conduct formal quality assessments of the included studies. To ensure comprehensive coverage, we supplemented our database searches with a review of reference lists from identified systematic reviews and relevant grey literature from global health and artificial intelligence organizations identified through targeted Google searches.

We anticipated substantial heterogeneity in study designs and evidence types. To account for this heterogeneity, we distinguished between individual-level capacity strengthening (i.e., training researchers to use AI tools for single-task research applications) and institutional or system-level capacity strengthening (i.e., efforts that build sustained research autonomy and infrastructure). This distinction informed how we grouped examples in the narrative synthesis.

After an initial review, we confirmed that two inclusion-eligible papers contained primary empirical data. The remaining six papers are descriptive reports or narrative reviews, for which there are no existing risk-of-bias instruments. For this reason, and because formal appraisal tools cannot be applied consistently across such varied study designs, we did not conduct a structured risk-of-bias or certainty assessment, and instead extracted and described methodological limitations narratively for transparency. Two reviewers extracted and discussed study-level methodological limitations (i.e., sampling approach, data origin, analytic transparency, and ethical reporting).

## Results

Our systematic review identified 305 papers, of which 8 met the inclusion criteria (Table 1). For the review of reviews, our search strategy identified 14 papers, of which 8 were included in the final analysis (Figs 1 and 2). Notably, the French, Portuguese, and Spanish searches on PubMed and Scopus yielded either no results or none that were relevant, whereas the SciELO searches yielded one relevant Spanish result. The PRISMA flow diagrams detail the screening process and reasons for exclusion at the full-text review and extraction phases for the English searches on PubMed and Scopus.

Papers meeting the inclusion criteria for the systematic review spanned several sectors, including environmental sciences (n = 1), academia (n = 1), evaluation of government systems (n = 1), and public health and biomedicine (n = 4). Four papers had a global scope, while two focused on the African continent, one on Mexico, and one on the Philippines. While we did not systematically classify author origin, the institutional addresses listed in several global papers indicate that many contributing teams were based in HIC institutions, reflecting the broader concentration of AI research efforts in these settings.

Three papers presented primary data on capacity-strengthening initiatives using AI: one utilized semi-structured interviews to evaluate global perspectives on AI for enhancing research capacity [19], the second used data gathered by an internal monitoring team to analyze an inter-institutional networking space to promote knowledge transfer on and use of AI in Mexico [20], and the third was a descriptive report on capacity-strengthening efforts in the Philippines [21]. The remaining publications consisted of commentaries or review articles. All papers were published in either 2023 or 2024.

Of the eight reviews identified for the review of reviews, the general focus was global health education (n = 4), sexual and reproductive health (n = 2), authorship in collaborative global health research (n = 1), and humanitarian health (n = 1). All papers were published after 2019, with three published in 2024. Determining whether included studies originated from HIC or LMIC research teams proved challenging because researchers from one setting may hold affiliations in institutions from the other. Systematically establishing author identity or institutional provenance was beyond the scope of this review.

**Table 1. Summary of the eight records included in the systematic review.**

| Lead author (year) | Geographic focus | Study type and data source | Primary objective | Methods and data | Relevant findings |
|---|---|---|---|---|---|
| Ahadi (2023) [31] | Global | Literature review | Map applications, challenges and opportunities of ChatGPT adoption in higher education | Broad search in conference proceedings; descriptive synthesis. | Identifies automation, sentiment analysis and NLP use-cases for AI in research. |
| Cadiz (2023) [21] | Philippines | Descriptive program report | Document national AI training series for universities and agencies | Overview of 5-day modular workshops; participant list and sample curriculum. | Demonstrates demand for GIS/ AI capacity building. |
| Chen (2024) [25] | Global | Literature review | Describe how ML/AI can advance life-course epidemiology | Synthesis of relevant literature and integration with life-course principles. | Outlines opportunities for improving life-course epidemiology, including causal inference. |
| Khan (2024) [19] | Global | Qualitative study (semi-structured interviews) | Explore perceived benefits and threats of ChatGPT for environmental Impact Assessment practice | Snowball recruitment; thematic analysis of interview transcripts. | Highlights time-savings potential. Notes concerns over data quality, plagiarism and reduced field work. |
| Kong (2023) [38] | Global South | Narrative viewpoint | Call for responsible, explainable, and local AI for clinical public health | Synthesis of relevant literature and review of reports submitted by partners to the ACADIC project. | Argues for equitable governance, data sovereignty and capacity investment to avoid widening North-South AI gap. |
| Nji (2024) [26] | African continent | Narrative viewpoint | Call for structural-biology capacity through BioStruct-Africa | Workshop description and stakeholder reflections. | Emphasizes training demand. Urges sustained funding and regional infrastructure. |
| Ramos (2023) [20] | Mexico | Descriptive case study | Analyze an alliance as a mechanism for building AI-related research capacities | Qualitative analysis drawing on interviews with project leads, internal monitoring reports, participation in alliance activities, and document review | Describes a "triple transfer" process—technology transfer to UNAM, capacity-building among research groups, and early-stage efforts to transfer outputs to vulnerable communities. |
| Turon (2024) [36] | African continent | Narrative viewpoint | Assess AI uptake in infectious disease labs and suggest LLM-based solutions | Experience from Ersilia Open-Source Initiative | Finds AI use rare. Proposes open-source LLM agents to lower entry barriers. |

We identified several ways AI can strengthen public health research capacity and notable gaps. Although several included studies describe AI being used for single research tasks rather than system-level capacity building, our analysis distinguishes between two complementary forms of capacity strengthening: training researchers to use task-specific AI tools and developing institutional capacities that support sustained, autonomous research activity. Both forms contribute to reducing dependence on external expertise, but they operate at different levels and yield different types of impact. The key identified themes include data analysis and research productivity, literature reviews and knowledge management, training and capacity strengthening, expanding access to methodological expertise, and writing support.

All eight records in the systematic review exhibited at least one serious methodological limitation; most papers did not employ rigorous study designs, while others did not provide sufficient information about the specific tools used or adapted for LMIC contexts. Therefore, each was judged to be at high risk of bias.

## Data analysis and research productivity

AI and other advanced analytic tools are used to analyze and interpret complex datasets. Studies in our review indicate that these technologies can facilitate data cleaning, pattern detection, and predictive modeling, which previously required extensive manual effort and statistical expertise. Further, AI tools can handle large volumes of structured and unstructured information. For example, though not identified in our review, the broader literature indicates that health researchers have

PLOS Digital Health

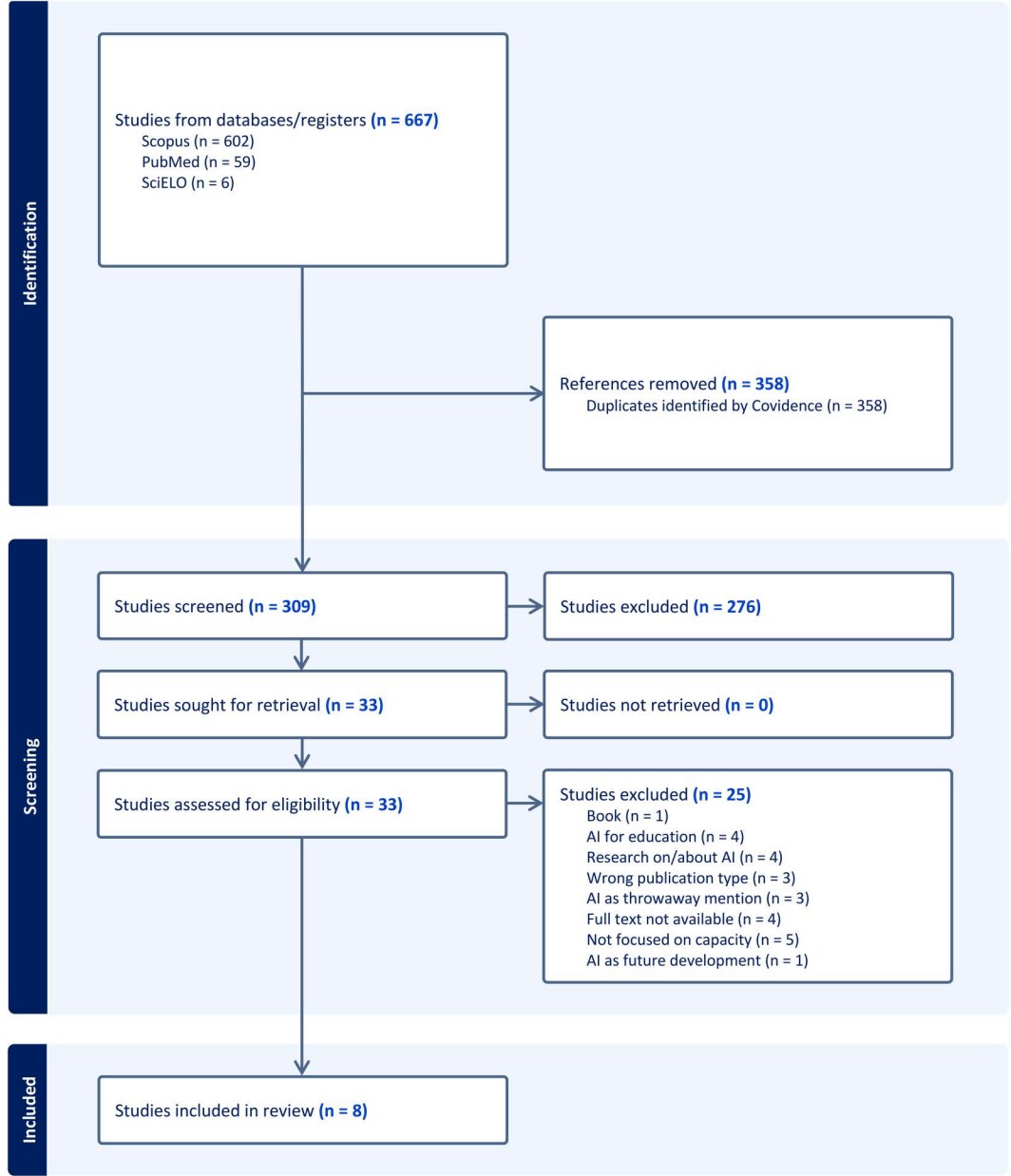

**Fig 1. PRISMA diagram for the systematic review on AI for capacity strengthening efforts.** The figure describes the identification process for the eight papers that were included in the systematic review focused on AI for capacity strengthening efforts. The search strategy targeted articles published in English, French, Portuguese, and Spanish.

used deep learning algorithms to analyze electronic medical records [22], genomics data [23], and real-time surveillance data [24]. The expansion of digitized health systems and remote sensing data in LMICs has created opportunities for AI to support research efforts in such settings.

In one paper identified from our systematic review, researchers examine the potential role of AI in strengthening research capabilities, specifically in life-course epidemiology [25]. The authors outline how AI and ML tools could expand

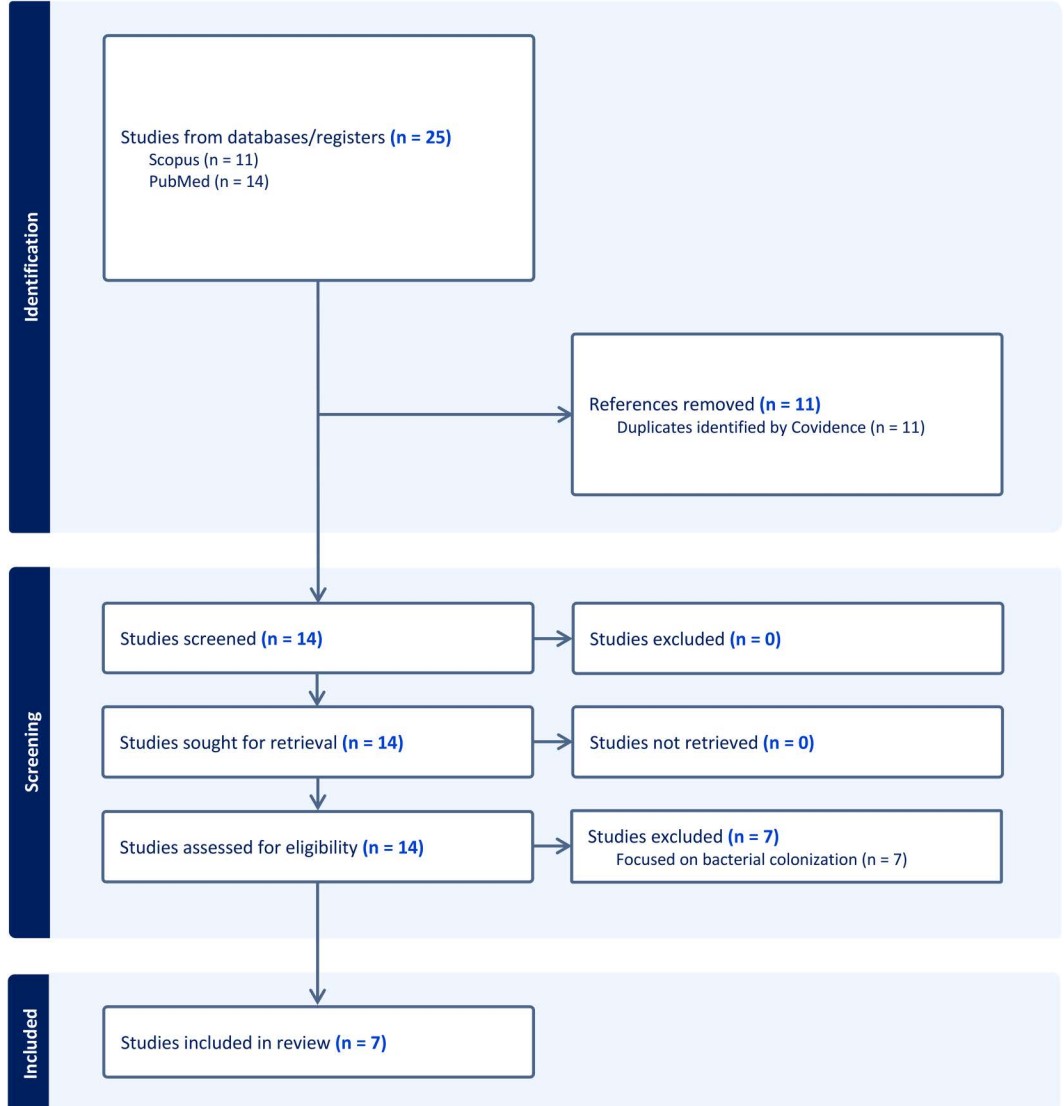

**Fig 2. PRISMA diagram for the review of reviews on decolonization of research capacity.** The figure describes the identification process for the seven papers that were included in the systematic review focused on decolonization of research capacity. The search strategy targeted articles published in English, French, Portuguese, and Spanish.

researchers' analytical capabilities by enabling them to process and analyze complex multimodal datasets that were previously challenging to handle with traditional methods. They suggest these technologies can democratize advanced analytical capabilities, allowing researchers with varying statistical expertise to conduct sophisticated longitudinal analyses. The authors propose an integrated framework that combines AI and ML tools specifically for life-course epidemiology and addresses how these technologies can enhance researchers' ability to identify public health interventions, analyze temporal patterns, and generate evidence-based insights for public health policies and programs.

In LMICs, the literature suggests that AI-driven analytics may benefit researchers with limited access to expert statisticians or high-performance computing centers [26]. We identified one paper in the supplemental review in which the authors describe efforts to expand the capacity for analyzing large epidemiological and genomic datasets within African

institutions [27]. According to this paper, open collaboration and resource sharing among local universities and global partners are important for African institutions to benefit from these emerging technologies. The authors suggest that efforts to build capacity could be enhanced by developing a comprehensive framework for data science health research governance across the continent. We also found evidence that in SRHR research, AI-powered data analysis tools have been used to track trends in contraceptive use [28] and maternal health outcomes in LMICs [29]. By leveraging AI-generated insights, researchers can design interventions that address unmet needs in reproductive health services.

Another initiative we identified through our review of reference lists and relevant grey literature is an open-source AI platform to support drug discovery research on neglected diseases in African and Latin American laboratories [27]. According to the authors, this project provides pre-trained ML models for tasks like predicting anti-malarial compound activity. It was designed to enable local scientists to run advanced computational experiments without developing their own algorithms. The authors also state that the platform aims to foster equitable research opportunities and boost collaborative innovation by lowering technical and financial barriers.

## Literature reviews and knowledge management

Four papers identified through the systematic review address how AI tools could be used for knowledge management activities. According to these papers, AI tools could help researchers sift through massive volumes of publications in a fraction of the time previously required. For example, the authors suggest that text-mining algorithms and large language models (LLMs) could facilitate the rapid identification of keywords, extraction of relevant abstracts, and categorization of articles by thematic areas [30]. The authors further suggest these capabilities could streamline the initial stages of evidence synthesis, potentially allowing investigators to focus on other aspects of the research.

One paper from the systematic review explores the integration of LLMs for literature curation and knowledge management [31]. The authors describe using such tools for automating tasks like information retrieval and theme extraction. They emphasize how such functions may improve research efficiency. They also highlight using LLMs to enhance scholarship by streamlining article identification and screening to offer preliminary summaries of relevant texts. The paper discusses the challenges (e.g., regulatory compliance and algorithmic bias) and the emerging opportunities (e.g., individualized learning experiences) associated with LLM adoption in research capacity development.

Several papers highlight that researchers need to remain aware of potential pitfalls when relying on AI for literature reviews and knowledge management. They note that LLMs can sometimes produce misleading summaries or "hallucinated" references if not tethered to verified datasets [32]. Another paper raises ethical questions about respecting intellectual property rights and avoiding unintentional plagiarism [33]. Last, authors from another study point out that many AI tools have been trained on data from high-income countries and, therefore, could overlook regional publications, potentially perpetuating knowledge inequities rather than reducing them [34].

## Training and capacity-strengthening efforts

Five papers in our review noted that developing the human skills required to use AI effectively appears crucial for realizing the full potential of advanced analytics in research [19]. According to these studies, even the most sophisticated AI tools demand contextual understanding, problem-solving abilities, and rigorous oversight to ensure ethical and accurate outcomes. The literature suggests that training programs and workshops may help bridge this gap by introducing researchers, students, and practitioners to best practices using these tools. One paper also analyses the value of an inter-institutional alliance to promote knowledge transfer from universities to other industries and sectors in Mexico. The authors of these papers propose that such initiatives could foster interdisciplinary collaboration, potentially enriching the research environment with diverse perspectives and expertise.

A descriptive capacity-strengthening report from the Philippines included in the review highlights the use of AI, alongside geographic Information systems (GIS) and radar remote sensing techniques [21]. According to the authors,

the training project focused on enhancing institutional skills and infrastructure by offering hands-on experience with cutting-edge technologies to address local environmental and public health challenges. The authors emphasize the necessity of interdisciplinary collaboration, where shared knowledge and joint problem-solving can drive effective AI implementation. They propose that by cultivating a new generation of skilled professionals, such initiatives may help develop sustainable research ecosystems that leverage technology appropriately.

We found specific examples of where SRHR research represents one area where AI-enabled capacity strengthening has gained traction. AI-driven natural language processing tools have been used to analyze policy documents and public health reports to assess gaps in family planning services [35]. In several LMICs, AI-powered platforms have facilitated training workshops designed for SRHR researchers, equipping them with skills to apply predictive analytics in sexual health epidemiology and intervention design.

In another example from our review, BioStruct-Africa is described as a grassroots initiative designed to empower African scientists through training and capacity building in structural biology [26]. According to the authors, the group promotes AI-driven tools like AlphaFold for protein structure prediction through hands-on workshops, mentoring programs, and collaborative networks. A related paper identified through reviewing the reference list underscores the importance of AI literacy among infectious disease researchers who need practical skills to align advanced computational models with day-to-day laboratory operations [36]. The authors of both papers suggest that tailored capacity-building programs can bridge the gap between cutting-edge AI technologies and local research contexts.

## Expanding access to methodological support

Several papers in our review suggest that generative AI tools and LLMs might serve as methodology consultants for researchers [19,31]. According to these authors, these systems could provide real-time research design, data analysis, and technical troubleshooting guidance. The literature proposes that they could expedite tasks such as statistical test selection, analytic strategy refinement, and clarification of best practices for large datasets. Some authors argue that AI-driven platforms can simulate expert knowledge and, thus, potentially help reduce the gap between novice and experienced researchers. Papers in our review indicate that researchers who engage in iterative dialogue with an LLM might identify potential pitfalls and enhance their methodological rigor.

A recent perspective on African infectious disease research highlights the potential of large language models to democratize access to advanced methodologies [36]. According to the paper, these systems could provide immediate, contextually relevant information to laboratories with limited resources. The authors assert that AI tools could enhance local research capabilities and help overcome infrastructure limitations [36]. The same paper notes that generative AI systems can still produce "hallucinations," making thorough verification essential. Two papers in our review suggest that when used as research consultants, the quality of LLMs depends on the researchers' knowledge and should not be seen as replacements for human expertise. Instead, generative AI could be used to supplement research capacity [19,31].

## Writing support

Two papers in our review suggest that generative AI tools could enhance scientific writing through automated drafting, grammar correction, and structural improvements [19,31]. According to these authors, these applications could enable researchers to focus on substantive work like data interpretation and conceptual development. The papers also note that the tools can help non-native English speakers navigate language barriers for writing proposals and peer-reviewed publications. The authors argue this would help expand access to high-quality editorial support, reducing disparities between researchers with institutional access to professional editing services and those without.

One article identified in the systematic review examined the role of an LLM in impact assessments. It emphasizes its utility in drafting reports, enhancing grammar, and minimizing the time and effort required for writing [19]. The authors emphasize that while AI-generated text can streamline the writing process, human oversight remains crucial to ensure

accuracy, contextual appropriateness, and adherence to ethical standards. They also highlight the importance of transparency in AI use, recommending researchers disclose when and how AI tools are employed to create written outputs. The authors suggest that with careful application and ongoing refinement, generative AI can be an invaluable partner in producing high-quality documents that effectively convey research findings and arguments.

### Review of reviews

Across the eight included reviews, a recurrent theme is a need to address power imbalances rooted in colonial legacies, with authors emphasizing local leadership and equitable partnerships in research design, implementation, and authorship. The reviews also underscore that inclusive, community-led initiatives and consistent, context-specific capacity building are critical to shifting structural barriers and advancing anti-colonial mindsets. They highlight that sustainable funding models and ethical frameworks attuned to local perspectives help ensure meaningful, collaborative engagements in generating global health knowledge.

## Discussion

We conducted complementary reviews to explore how AI can contribute to the movement to shift power in global health from a knowledge production perspective. Both the systematic review and the review of reviews revealed a relatively small but growing body of literature identifying the relevance of AI-driven tools and methods in capacity-building initiatives. Studies highlighted promising practices for advanced data analysis, knowledge management, and writing support. Most published examples address individual-level capacity strengthening through training on task-specific AI tools, whereas evidence of institutional or system-wide improvements remains limited. These forms of capacity strengthening cannot substitute for reforms to the structural forces that shape global research priorities, funding, and authorship norms.

A key finding across multiple papers was AI's potential to address significant skill and resource deficits in research settings. In contexts where statistical expertise, computational infrastructure, or editorial support are scarce, AI tools could help bridge these gaps, allowing researchers to undertake projects that would otherwise be infeasible. However, the literature cautions that technology alone cannot substitute for fundamental capacity building and infrastructure development.

However, the reviews also revealed that the development and deployment of AI tools in global health research remain uneven, with much of AI design and use still concentrated in high-income contexts. While the literature belies an enthusiasm for the democratizing and decolonizing potential of AI in global health research, many authors are aware of the risks that accompany the use of relatively new technologies, particularly those that are heavily shaped by a limited context yet poised to be applied to a variety of sectors and settings. Although we did not undertake a formal assessment of author provenance, the institutional affiliations reported in many included papers suggest that HIC-based institutions continue to play a leading role in AI-related research, underscoring the structural asymmetries that shape knowledge production.

Concerns about quality, transparency, privacy, and accountability abound, especially in settings where limited infrastructure can lead to constraints in mitigating these concerns, whether through limited personnel, familiarity, or access. While novel applications such as LLMs may democratize access to advanced methods, further empirical evidence about their effectiveness in strengthening research capacity is required. The literature suggests that AI must be seen as a tool to supplement, rather than supplant, researchers' expertise and capacities.

Integrating AI into SRHR research capacity-building aligns with broader efforts to decolonize global health. By equipping local researchers with AI tools to analyze gender disparities in healthcare access, identify policy gaps, and strengthen advocacy efforts, AI can help shift power dynamics toward more locally led and context-driven solutions for many domains within global health, including SRHR.

Several papers in our review suggest that AI-driven tools for supporting research could empower all researchers, including those in LMICs, to tackle data-heavy projects that might have been impractical otherwise. The literature emphasizes that the success of these AI applications depends on data quality and context appropriateness, as biased or

incomplete datasets can mislead AI. Researchers have emphasized the need to maintain data integrity and understand algorithmic output limitations, ensuring that AI's benefits are realized without compromising scientific rigor [37].

The findings from our review suggest that targeted training programs that integrate AI components into curricula and workshops to develop local expertise and strengthen research capacity could be highly valuable. Based on successful examples identified in our review, effective training initiatives combine hands-on technical skill development with the conceptual understanding of AI capabilities and limitations. Several papers propose that collaborative initiatives that bring together governments, academic institutions, and international donors to support digital infrastructure, data governance frameworks, and mentorship programs could serve as the underpinning foundations of such programs, led by interdisciplinary teams that bring together expertise in computer science, public health, and community engagement.

Authors in our review suggest that capacity-strengthening initiatives could enhance AI's potential impact in decolonizing global health by increasing AI-powered tools to dismantle colonial legacies in knowledge production by promoting local leadership in research design and execution. However, several papers caution that AI technologies risk reinforcing existing power imbalances and perpetuating colonial dynamics without careful implementation and local ownership. If AI tools are developed primarily in high-income settings with minimal input from LMIC researchers or embed cultural assumptions and epistemological biases from dominant knowledge systems, they may inadvertently entrench rather than challenge colonial mindsets in research.

Our review found limited empirical evaluation of AI-focused capacity-strengthening initiatives. Future research could benefit from developing context-specific metrics that assess technical skill development and measures of research independence, local leadership, and contributions to addressing locally prioritized health challenges. Longitudinal studies tracking how AI tools influence research output, collaboration patterns, and knowledge translation in different settings would provide valuable insights beyond immediate training outcomes.

Our review identified several risks associated with AI adoption in research capacity strengthening. Multiple papers highlighted concerns about data privacy, algorithmic bias, and the potential for AI to amplify existing inequities if not thoughtfully implemented. Authors noted that AI systems trained primarily on data from high-income countries might perform poorly when applied to different contexts or populations. There were also concerns about dependency on proprietary AI systems controlled by a few technology companies, potentially creating new forms of technological colonialism. The literature emphasizes that acknowledging and actively mitigating these risks through inclusive governance and thoughtful implementation is essential for AI to contribute positively to research capacity strengthening and decolonization efforts.

Our findings indicate promising practices and practical next steps, but we acknowledge three limitations in the review. Firstly, the primary working language of both reviewers is English, and while efforts were made to conduct searches in French, Portuguese, and Spanish, we acknowledge that lack of fluency in all three languages and familiarity with their respective literature bases likely led to the exclusion of local innovations, promising practices, and next steps developed in various non-Anglophone contexts. Secondly, AI is developing at an exponential pace incompatible with the relatively stretched-out timelines of peer-reviewed research, leaving more recent developments absent from the literature while causing some research findings to become quickly outdated and thus limiting the applicability of some reviewed evidence. Finally, the limited number of articles meeting the inclusion criteria, which is partly a reflection of the timelines in peer-reviewed publishing, constrains the generalisability of the conclusions.

These reviews demonstrate the potential for AI to transform research capacity in LMICs by democratizing access to advanced analytical tools, providing methodological support, and helping overcome resource limitations that have historically restricted research opportunities. However, without timely efforts focused on expanding equitable governance and local leadership, the gap in research capacity between LMICs and HICs could widen further, perpetuating the power asymmetries that current efforts seek to dismantle. Instead, by centering local leadership and knowledge systems in the development and implementation of these technologies, AI could become a powerful ally in creating a more equitable global research ecosystem that amplifies diverse voices and perspectives in the production of health knowledge.

## Supporting information

**S1 Text. Supplemental material.**
(DOCX)

**S1 Checklist. PRISMA Checklist.**
(DOCX)

## Author contributions

**Conceptualization:** Brian Wahl, Daniel D. Redpath, Pascale Allotey, Nina Schwalbe.

**Data curation:** Brian Wahl, Tiffany Nassiri-Ansari.

**Formal analysis:** Brian Wahl, Tiffany Nassiri-Ansari, Pascale Allotey, Nina Schwalbe.

**Funding acquisition:** Pascale Allotey, Nina Schwalbe.

**Investigation:** Brian Wahl, Tiffany Nassiri-Ansari.

**Methodology:** Brian Wahl, Tiffany Nassiri-Ansari, Pascale Allotey.

**Supervision:** Brian Wahl, Nina Schwalbe.

**Writing – original draft:** Brian Wahl, Tiffany Nassiri-Ansari, Nina Schwalbe.

**Writing – review & editing:** Brian Wahl, Tiffany Nassiri-Ansari, Daniel D. Redpath, Pascale Allotey, Nina Schwalbe.

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
