## [Decision Letter · Decision Letter 0]

5 Aug 2025

Response to Reviewers'. This file does not need to include responses to any formatting updates and technical items listed in the 'Journal Requirements' section below.'. This file does not need to include responses to any formatting updates and technical items listed in the 'Journal Requirements' section below.* A marked-up copy of your manuscript that highlights changes made to the original version. You should upload this as a separate file labeled 'Revised Manuscript with Track Changes'.'.* An unmarked version of your revised paper without tracked changes. You should upload this as a separate file labeled 'Manuscript'.'. If you would like to make changes to your financial disclosure, competing interests statement, or data availability statement, please make these updates within the submission form at the time of resubmission. Guidelines for resubmitting your figure files are available below the reviewer comments at the end of this letter. We look forward to receiving your revised manuscript. Kind regards, Charles B. DelahuntAcademic EditorPLOS Digital Health Charles DelahuntAcademic EditorPLOS Digital Health Leo Anthony CeliEditor-in-ChiefPLOS Digital Healthorcid.org/0000-0001-6712-6626 **Journal Requirements:** If the reviewer comments include a recommendation to cite specific previously published works, please review and evaluate these publications to determine whether they are relevant and should be cited. There is no requirement to cite these works unless the editor has indicated otherwise.  **Additional Editor Comments (if provided):** Good morning, and thank you for your patience (if applicable) while we rounded up reviews. The good news is that the reviews are well-informed and thoughtful.

Both reviewers raise important points, and I urge you to fully address them, even at the cost of some effort.

Other comments:

The restriction of the HIC-only, English-only search scope, raised by both reviewers, is ironic and somewhat meta, given the topic. Addressing this by expanding your search space to include non-HIC sources would substantially increase the credibility and value of the paper. I suggest that you enlist Spanish, French, and maybe Portuguese speakers to search and navigate non-English literature. For Spanish literature, please see this AAAS article: https://www.science.org/content/article/latin-american-journals-are-open-access-pioneers-now-they-need-audience . I believe SciELO is a good source.

Re the distinction between isolated, single task uses of AI vs capacity-building uses of AI: Some of the AI uses you cite (eg lines 184-185, 208 - 209) appear to be cases where AI is used to tackle a particular research topic, which is a different thing from more general use to increase capacity. How does this affect your assessment?

  ?>**Reviewers' Comments:** Reviewer's Responses to Questions

**Comments to the Author**

1. Does this manuscript meet PLOS Digital Health’s publication criteria? Is the manuscript technically sound, and do the data support the conclusions? The manuscript must describe methodologically and ethically rigorous research with conclusions that are appropriately drawn based on the data presented.? Is the manuscript technically sound, and do the data support the conclusions? The manuscript must describe methodologically and ethically rigorous research with conclusions that are appropriately drawn based on the data presented.

Reviewer #1: No

Reviewer #2: Yes

2. Has the statistical analysis been performed appropriately and rigorously?

Reviewer #1: No

Reviewer #2: N/A

3. Have the authors made all data underlying the findings in their manuscript fully available (please refer to the Data Availability Statement at the start of the manuscript PDF file)?

The PLOS Data policy requires authors to make all data underlying the findings described in their manuscript fully available without restriction, with rare exception. The data should be provided as part of the manuscript or its supporting information, or deposited to a public repository. For example, in addition to summary statistics, the data points behind means, medians and variance measures should be available. If there are restrictions on publicly sharing data—e.g. participant privacy or use of data from a third party—those must be specified.requires authors to make all data underlying the findings described in their manuscript fully available without restriction, with rare exception. The data should be provided as part of the manuscript or its supporting information, or deposited to a public repository. For example, in addition to summary statistics, the data points behind means, medians and variance measures should be available. If there are restrictions on publicly sharing data—e.g. participant privacy or use of data from a third party—those must be specified.

Reviewer #1: Yes

Reviewer #2: No

4. Is the manuscript presented in an intelligible fashion and written in standard English?

Reviewer #1: Yes

Reviewer #2: Yes

Reviewer #1: The manuscript addresses an important and timely topic at the intersection of artificial intelligence (AI), global health (GH), and research equity. However, the central theoretical framing requires significant clarification and revision.

Major Comments:

1. Theoretical Inconsistencies and Lack of Clarity:

The manuscript begins by identifying a gap in agenda-setting and knowledge production and posits that AI may help build research capacity in low- and middle-income countries (LMICs), thereby shifting power dynamics in global health. However, the literature review suggests that these power imbalances are primarily due to the dominance of institutions in high-income countries (HICs) in setting research priorities, directing funding, and defining what counts as “valid” knowledge. The manuscript then pivots back to promoting capacity building via AI—without sufficiently addressing that the core issue may not be capacity gaps in LMICs, but rather inequities in funding flows and decision-making authority. This disconnect in theoretical framing weakens the central argument. The authors should reconcile these perspectives or reframe the theory to accurately reflect the root causes of power asymmetries in global health.

2. Scope of the Systematic Review:

The papers included in the systematic review are not all focused on LMICs. Of the three global papers cited, two originate from institutions based in HICs, and the third (Khan, 2023) includes a mixture of institutions, though the specific contributions of Northern versus Southern actors are unclear. This distinction matters, as the paper draws conclusions about the role of AI in LMICs based on existing infrastructures, which are often shaped by Northern-led institutions. Greater transparency is needed regarding the geographical and institutional origins of the reviewed studies, especially if conclusions are being generalized to LMIC contexts.

3. Framing and Methodological Concerns:

The manuscript must make a clearer choice in framing: is it exploring what large language models (LLMs) can do to assist researchers in LMICs, or is it proposing a broader theory of AI-led shifts in power dynamics in global health? The current combination of both makes the theoretical and methodological structure feel underdeveloped. An additional "Review of Reviews" appears forced and isn’t adding much value to the overall theory.

Minor Comments:

# Lines 72–75: The reference to SRHR appears abruptly. Consider explaining how this focus emerged or provide a clearer transition.

# Lines 161–167: Please include relevant citations to support the claims made.

# Lines 174–175: The mention of "serious methodological limitations" is vague. Specify what these limitations are.

Reviewer #2: I would like to congratulate the authors on this review which tackles an important, under-explored question—how AI could rebalance research capacity and power between high- and LMIC. The dual-synthesis design (systematic review of primary studies plus umbrella review of de-colonisation literature) is conceptually strong, and of interest and the supplementary material now supplies full search strings and clear PRISMA flow charts. Nevertheless, several elements remain incomplete, limiting a little bit some of the methodological rigour and reproducibility.

First, no protocol was registered and none seems to be provided. PROSPERO registration is not strictly mandatory for publication, but it is considered best practice and from what this reviewer can tell it is encouraged by PLOS to protect against post-hoc decision-making; perhaps the authors should register retrospectively (or upload a dated protocol) and document any changes?

Second, risk-of-bias and certainty assessments are entirely absent. The justification by the authors seems solid but even with heterogeneous study designs, the primary record should be appraised with an appropriate JBI/NIH tool, and the eight included reviews assessed with ROBIS or AMSTAR-2; overall certainty could then be graded with GRADE-CERQual. Without this step, the narrative synthesis rests on un-weighted evidence and its conclusions may over-reach. That being said, this reviewer is not an expert on systematic reviews and perhaps this is not disqualifying given the authors provided justification.

Third, the search remains limited to PubMed and Scopus, in English only. Extending to EMBASE, Web of Science, Global Health, IEEE Xplore and at least one regional database, plus a grey-literature/pre-print sweep, would reduce selection bias and better capture LMIC scholarship. Any language restriction must be justified.

Fourth, the Data-Availability statement says “NA”, which could contravene PLOS policy. De-duplicated citation lists, screening logs, extraction sheets and any analytic code should be deposited in an open repository (e.g., Zenodo or OSF) and the DOI cited.

Finally, minor editorial issues persist (line-numbers in the text, “World Count” header, duplicated Chen 2024 reference).

In sum, the manuscript is topical and potentially impactful, but it requires some revision to meet PLOS’ standards for transparency and methodological soundness as far as systematic reviews go. Addressing the four points above—protocol registration, risk-of-bias certainty appraisal (potentially) , expanded search strategy, and compliant data sharing—together with routine polishing, will substantially strengthen the work and lend credibility to its claims.

**Do you want your identity to be public for this peer review?** For information about this choice, including consent withdrawal, please see our Privacy Policy..

Reviewer #1: No

Reviewer #2: No

**Figure resubmission:** While revising your submission, please upload your figure files to the Preflight Analysis and Conversion Engine (PACE) digital diagnostic tool, https://pacev2.apexcovantage.com/. PACE helps ensure that figures meet PLOS requirements. To use PACE, you must first register as a user. Registration is free. Then, login and navigate to the UPLOAD tab, where you will find detailed instructions on how to use the tool. If you encounter any issues or have any questions when using PACE, please email PLOS at figures@plos.org. Please note that Supporting Information files do not need this step. If there are other versions of figure files still present in your submission file inventory at resubmission, please replace them with the PACE-processed versions. **Reproducibility:** To enhance the reproducibility of your results, we recommend that authors of applicable studies deposit laboratory protocols in protocols.io, where a protocol can be assigned its own identifier (DOI) such that it can be cited independently in the future. Additionally, PLOS ONE offers an option to publish peer-reviewed clinical study protocols. Read more information on sharing protocols at https://plos.org/protocols?utm_medium=editorial-email&utm_source=authorletters&utm_campaign=protocols To enhance the reproducibility of your results, we recommend that authors of applicable studies deposit laboratory protocols in protocols.io, where a protocol can be assigned its own identifier (DOI) such that it can be cited independently in the future. Additionally, PLOS ONE offers an option to publish peer-reviewed clinical study protocols. Read more information on sharing protocols at https://plos.org/protocols?utm_medium=editorial-email&utm_source=authorletters&utm_campaign=protocols

---

## [Decision Letter · Decision Letter 1]

26 Feb 2026

Artificial intelligence for research capacity strengthening: Two reviews and a pathway to shift power in global health

PDIG-D-25-00220R1

Dear Dr. Wahl,

We are pleased to inform you that your manuscript 'Artificial intelligence for research capacity strengthening: Two reviews and a pathway to shift power in global health' has been provisionally accepted for publication in PLOS Digital Health.

Best regards,

Charles B. Delahunt

Academic Editor

PLOS Digital Health

**Additional Editor Comments (if provided):**

Good morning -

Thank you for your edits,which sufficiently addressed reviewers' comments. Thank you also for your patience with a more-drawn-out-than-ideal process.

A couple minor points:

- line 129 - 130: The same authors covered both topics (just confirming that this is correct).

- line 399: "belies" may not be the correct word here.

**Reviewer Comments (if any, and for reference):**

Reviewer's Responses to Questions

**Comments to the Author**

Reviewer #1: All comments have been addressed

publication criteria? Is the manuscript technically sound, and do the data support the conclusions? The manuscript must describe methodologically and ethically rigorous research with conclusions that are appropriately drawn based on the data presented.? Is the manuscript technically sound, and do the data support the conclusions? The manuscript must describe methodologically and ethically rigorous research with conclusions that are appropriately drawn based on the data presented.

Reviewer #1: Yes

3. Has the statistical analysis been performed appropriately and rigorously?

Reviewer #1: N/A

4. Have the authors made all data underlying the findings in their manuscript fully available (please refer to the Data Availability Statement at the start of the manuscript PDF file)?

The PLOS Data policy requires authors to make all data underlying the findings described in their manuscript fully available without restriction, with rare exception. The data should be provided as part of the manuscript or its supporting information, or deposited to a public repository. For example, in addition to summary statistics, the data points behind means, medians and variance measures should be available. If there are restrictions on publicly sharing data—e.g. participant privacy or use of data from a third party—those must be specified.requires authors to make all data underlying the findings described in their manuscript fully available without restriction, with rare exception. The data should be provided as part of the manuscript or its supporting information, or deposited to a public repository. For example, in addition to summary statistics, the data points behind means, medians and variance measures should be available. If there are restrictions on publicly sharing data—e.g. participant privacy or use of data from a third party—those must be specified.

Reviewer #1: Yes

5. Is the manuscript presented in an intelligible fashion and written in standard English?

Reviewer #1: Yes

Reviewer #1: Thank you for addressing the comments. Congratulations again on the important contribution.

**Do you want your identity to be public for this peer review?** For information about this choice, including consent withdrawal, please see our Privacy Policy..

Reviewer #1: No
